# A core outcome set for research and clinical practice in women with pelvic girdle pain: PGP-COS

Alexandria Remus[1]*, Valerie Smith[1], Annelie Gutke[2], Juan Jose Saldaña Mena[3], Siv Mørkved[4], Lena Nilsson Wikmar[5,6], Birgitta Öberg[7], Christina Olsson[6], Hilde Stendal Robinson[8], Britt Stuge[9], Francesca Wuytack[1]

1 School of Nursing & Midwifery, Trinity College Dublin, Dublin, Ireland, 2 Department of Health and Rehabilitation, Institute of Neuroscience and Physiology, University of Gothenburg, Göteborg, Sweden, 3 Universidad Estatal del Valle de Ecatepec, Ecatepec de Morelos, Mexico, 4 Department of Public Health and Nursing, Norwegian University of Science and Technology, Trondheim, Norway, 5 Division of Physiotherapy, Department of Neurobiology, Care Sciences and Society, Karolinska Institutet, Stockholm, Sweden, 6 Academic Primary Healthcare Centre, Stockholm, Sweden, 7 Department of Medicine and Health, Linköping University, Linköping, Sweden, 8 Department of Interdisciplinary Health Sciences, Institute of Health and Society, University of Oslo, Oslo, Norway, 9 Division for Neuroscience and Musculoskeletal Medicine, Oslo University Hospital, Oslo, Norway

* alexandria.remus2@gmail.com

**Data Availability Statement:** All relevant data are within the manuscript and its Supporting information files.

## Abstract

### Background

Inconsistent reporting of outcomes in clinical trials of women with Pelvic Girdle Pain (PGP) hinders comparison of findings and the reliability of evidence synthesis. A core outcome set (COS) can address this issue as it defines a minimum set of outcomes that should be reported in all clinical trials on the condition. The aim of this study was to develop a consensus-based COS for evaluating the effectiveness of interventions in PGP during pregnancy and postpartum for use in research and clinical practice.

### Methods

A systematic review of previous studies on PGP and semi-structured interviews with women were undertaken to identify all outcomes that were reported in prior studies and that are relevant to those experiencing the condition. Key stakeholders (clinicians, researchers, service providers/policy makers and individuals with PGP) then rated the importance of these outcomes for including in a preliminary PGP-COS using a 3-round Delphi study. The final COS was agreed at a face-to-face consensus meeting.

### Results

Consensus was achieved on five outcomes for inclusion in the final PGP-COS. All outcomes are grouped under the "life impact" domain and include: pain frequency, pain intensity/ severity, function/disability/activity limitation, health-related quality of life and fear avoidance.

**Funding:** We have received funding from the Belgian Chiropractic Union Research Fund (www. chiropraxie.org) and from the European Centre for Chiropractic Research Excellence (nikkb.dk/eccre). The funders had no role in the study design, data collection, management, data interpretation, report writing and decision to submit for publication.

**Competing interests:** The authors have declared that no competing interests exist.

## Conclusion

This study identified a COS for evaluating the effectiveness of interventions in pregnancy-related and postpartum-related PGP in research and clinical settings. It is advocated that all trials, other non-randomised studies and clinicians in this area use this COS by reporting these outcomes as a minimum. This will ensure the reporting of meaningful outcomes and will enable the findings of future studies to be compared and combined. Future work will determine how to measure the outcomes of the PGP-COS.

## Core outcome set registration

This PGP-COS was registered with COMET (Core Outcome Measures for Effectiveness Trials) in January 2017 (http://www.comet-initiative.org/studies/details/958).

## Introduction

Pelvic Girdle Pain (PGP) during and after pregnancy is a common complaint reported by women worldwide. It affects up to two thirds of women at some point during pregnancy and can persist postpartum [1–7]. PGP is a significant cause of disability, negatively affects quality of life and is one of the leading contributors to employee absenteeism during pregnancy [8–14]. The effects of PGP have a large economic, social and psychological impact on individual families and society, resulting in an urgent need for effective interventions worldwide. Although various interventions for the prevention and treatment of PGP have been studied, the resulting evidence is difficult to interpret due to, in part, the large variety of outcomes reported across studies [15, 16]. For example, a systematic review on physiotherapy modalities including 58 articles could not perform any meta-analysis due to heterogeneity across studies [15]. The inability to meta-analyse outcome data results in relying on evidence from smaller individual studies that provide lesser quality evidence to identify effective interventions. Additionally, recent work from Wutack and O'Donovan identified 46 different outcomes measured across 107 intervention studies or systematic reviews of interventions for PGP [17]. They also identified that different studies often use a variety of measurement instruments to capture the same outcomes. This heterogeneity in reported outcome measures is problematic not only for direct comparison between studies, but it also limits the ability for the aggregation of data across trials, which renders the translation to clinical practice difficult and sometimes impossible [18]. The ability to pool data and compare across studies would allow for robust meta-analyses which, in turn, will aid in determining the most effective interventions for PGP.

One approach to overcome the lack of uniformity in reported outcome measures is to develop a core outcome set (COS). A COS is a standardised set of outcomes which should be measured and reported, as a minimum, in all studies for a specific health area or condition [19]. A COS allows for findings to be combined, compared and contrasted, reduces potential for reporting bias and ensures that the data are useful and usable, but does not restrict researchers from measuring additional outcomes at their discretion. A PGP-COS would assist in promoting the health and well-being of women with pregnancy and postpartum related PGP through consistent and relevant outcome reporting worldwide.

The aim of our study was to develop a consensus-based COS for PGP during pregnancy and postpartum at should be used, as a minimum, for use in research and clinical practice.

## Methods

The study was prospectively registered in the COMET (Core Outcome Measures in Effectiveness Trials Initiative) database (Registration number: 958; http://www.comet-initiative.org/studies/details/958) and a detailed protocol was published [18]. The complete PGP-COS study involves five phases (Fig 1). Phase 1 [17], systematic review, has already been reported and phase 4 and 5 are in progress. In this paper, we focus on reporting the conduct and results of phases 2 and 3 and adhere to the Core Outcome Sets-STAndards for Reporting (COS-STAR) criteria and COMET guidance [20, 21]. Ethical approval for the study was granted from the lead researcher's University's Research Ethics Committee.

### Phase 1: Preliminary list of outcome measures

Phase 1 resulted in identifying 53 outcomes for use in phase 2, the Delphi study. Of these outcomes, 45 were identified in the systematic review [17] and 8 were identified via 15 interviews with women who experienced PGP during pregnancy or postpartum from three countries; Ireland (n = 5), Sweden (n = 5) and Mexico (n = 5). These women were recruited via physiotherapy and chiropractic clinics and provided written informed consent for interviews. Following review, these outcomes were grouped by outcome domain using the OMERACT filter 2.0 framework [22] (Table 1) and forwarded for use in phase 2. It is important to note that domain classification for potential outcomes identified in the literature is consistent with what was reported during phase 1 of the PGP-COS development [17].

### Phase 2: An international Delphi study

**Participant selection and recruitment.** A 3-round online Delphi study was used to reach consensus on the preliminary COS. Participants were recruited, internationally, from five PGP stakeholder groups: (1) clinicians, (2) patient representatives, (3) researchers, (4) researchers also working clinically and (5) policy makers and service providers. Potential participants were invited through mass invitational email (researchers identified from PubMed), professional organisations (such as world chiropractic, physiotherapy, surgical, osteopathic, patient, etc. groups) and social media (Facebook and Twitter). Snowball sampling was further encouraged. Invitees were provided with access to our online Participant Information Leaflet (PIL) which outlined the need for the study, the principles of a COS and what taking part in the study would involve. Access to the Delphi survey link was also provided as part of the invitation.

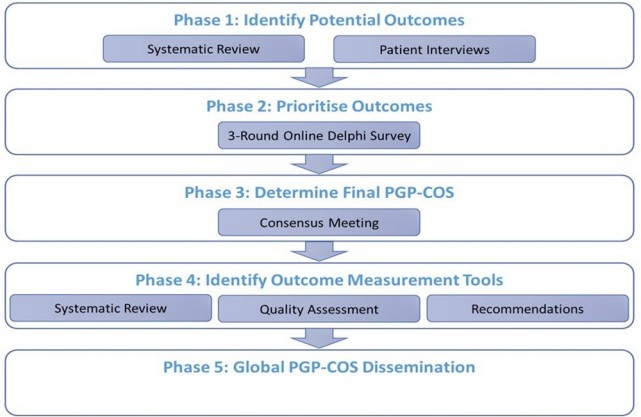

**Fig 1. PGP-COS study phases.**

**Table 1. Preliminary outcomes.**

| Life Impact | Economic Impact & Resource-Use Impact | Pathophysiological Manifestations & Clinical Tests | Adverse Events |
|---|---|---|---|
| *n = 30* | *n = 5* | *n = 16* | *n = 2* |
| Pain behaviour | Work ability | Anthropomorphic outcomes (BMI, height, weight, etc.) | Maternal adverse events/undesirable side effects |
| Pain character/type | Work performance | Body flexibility | Unborn/born child adverse events/ undesirable side effects |
| Pain frequency | Analgesia use | Functional mobility | |
| Pain intensity/severity | Cost | Gait endurance | |
| Pain location | Healthcare utilisation | Gait speed | |
| Full pain recovery | | Maternal pregnancy outcomes | |
| Function/disability/activity limitation | | Muscle endurance | |
| Physical activity levels/exercise limitations | | Muscle strength | |
| Need for a mobility aid | | New born outcomes | |
| Perceived body imbalance | | Outcomes from pain provocation/location tests | |
| Sexual functioning | | Posture | |
| Health related quality of life | | Pubis Symphysis mobility | |
| Health status | | Recovery of symptoms | |
| Family life impact | | Step length | |
| Social life impact | | Surgical outcomes | |
| Patient expectation of treatment | | Urinary Incontinence | |
| Patient satisfaction with life | | | |
| Patient satisfaction with treatment | | | |
| Anxiety | | | |
| Confidence | | | |
| Depression | | | |
| Dependence on others | | | |
| Emotional symptoms | | | |
| Fear avoidance | | | |
| Frustration | | | |
| Pain catastrophizing | | | |
| Self-efficacy | | | |
| Well-being | | | |
| Fatigue | | | |
| Sleep function | | | |

Only potential participants that provided informed consent in the appropriate consent section were able to proceed with the Delphi survey. Study participants were pseudonymised and blinded to all other study members and participants by one researcher who managed this process (AR).

**Delphi procedure.** The surveys for each of the Delphi rounds were created using Google forms [23]. Each round was open for 21 days. During the second and third rounds, a reminder email was sent to all non-responders seven days before the survey closed. Non-responders after each round closed were not invited to the subsequent round. Participants were sent a copy of their survey responses after each round for reference in completing the subsequent round.

**Table 2. Rating scales.**

| 5-point scale | | 9-point scale | |
|---|---|---|---|
| 1–2 | Not important | 1–3 | Not important |
| 3 | Unsure of importance | 4–6 | Unsure of importance |
| 4–5 | Important | 7–9 | Important |

The round 1 survey presented the 53 outcomes identified in phase 1 (Table 1). A lay definition and/or examples were provided for outcomes where deemed necessary (S1 Table). Participants were asked to rate the importance of each outcome for inclusion in a PGP-COS using one of two rating scales; 5-point scale or a 9-point scale (Table 2), which the participants were randomly allocated to when they clicked the survey link. The use of these two scales was included as part of an embedded methodological study comparing the potential effect of different scales on the final outcomes in a COS development [18]. This embedded study was included in the development of the PGP-COS to explore the impact of different rating scales on the final COS. There is currently no consensus on which rating scale should be used in COS development and, as such, different scales have been used in previous COS. The results of the embedded methodological study are reported elsewhere [24]. During the Delphi study, however, any outcome rated as important on either scale (Table 2), as per our consensus definition detailed below, was forwarded to the subsequent round. During round 1, participants were also given the opportunity to add up to three outcomes not identified in the preliminary list that they believed were important to be included in the COS using free-text responses.

In addition to their individual round 1 results, participants in round 2 were presented with the proportion of participants in each stakeholder group who rated each outcome as "important". Participants were then asked to re-rate the importance of each outcome for inclusion in the PGP-COS using the same scale they were randomly allocated in round 1 (5-point or 9-point) and to rate the importance of any new outcomes added following round 1 (S1 Table).

In accordance with the COMET guidelines, outcomes forwarded from round 2 to round 3 were determined a priori based on consensus defined as ≥70% participants rating the outcome as "important" by three of the five stakeholder groups, one of which must have been the patient representative group [21, 25]. In round 3, participants were presented with the results from round 2, and were asked to re-rate the importance of each outcome for inclusion in the final PGP-COS.

Following round 3, outcomes were classified as "consensus in" (reaching a priori consensus) or "no consensus" (anything else). All outcomes that reached "consensus in" on the 5- and 9-point rating scales were combined to form the list of outcomes for discussion at the face-to-face consensus meeting.

## Consensus meeting

Agreement on the final PGP-COS was achieved at a 1-day face-to-face consensus meeting held on October 27th, 2019, in Antwerp, Belgium. The meeting was chaired by VS and led by AR, neither of whom had a vote on the outcomes at the meeting. Participants were eligible to attend the consensus meeting if they completed all 3 rounds of the Delphi survey and indicated an interest in attending the meeting by ticking a box on the round 3 survey. To ensure representation across stakeholder groups, Delphi study participants who indicated their interest in attending were sorted by stakeholder group and then randomly chosen to attend using a random number generator. International representation was also monitored as part of this process and was adequately achieved.

During the consensus meeting participants were presented with the list of outcomes resulting from the Delphi study. Each outcome, in turn, was individually considered by the consensus panel, and fully discussed. Following discussion, the panel members voted on the critical importance of including the outcome measure in the final PGP-COS. A priori consensus for the face-to-face consensus meeting was determined as $\geq$ 70% of all members voting "yes" for the inclusion of an outcome measure in the final PGP-COS. Voting was anonymous using smart technology and the Poll Everywhere (www.polleverywhere.com). At the end of voting, outcome measures that reached "consensus in" were presented to the consensus meeting panel for final consideration and agreement on inclusion in the final PGP-COS.

## Results

### Delphi study

Participant demographics for all three Delphi rounds are presented in Table 3. Overall, 205 stakeholders from 32 countries completed round 1. Of these, 147 (72%) completed round 2 and, of these, 132 (90%) completed round 3. This provided an overall Delphi retention rate of 64%.

The proportion of participants in each stakeholder group that rated each outcome as "important" on the 5-point and 9-point rating scales in rounds 1 to 3 is presented in S2 Table and the findings of the face-to-face consensus meeting are presented in Table 4. The stakeholder ratings are presented in full detail in S2 Table and Table 4 so that the outcomes can be tracked throughout the whole consensus process so comparisons can be made for full transparency. After adjusting for consensus in round 3, 25 outcomes were taken forward for discussion and voting at the face-to-face consensus meeting (Table 4).

### Consensus meeting

Twenty-five participants were invited to take part in the face-to-face consensus meeting. Thirteen participants agreed to participate; however, due to last minute travel emergencies, the final number participating in the meeting was 11 stakeholders. Consensus meeting participant characteristics, including multiple stakeholder group affiliations, are provided in Table 5. The results of the voting are presented in Table 4. Five outcome measures, all of which are included in the 'life impact' domain, achieved consensus and constitute the agreed final PGP-COS. These outcomes are pain frequency, pain intensity/severity, function/disability/activity limitation, health related quality of life and fear avoidance.

## Discussion

This study provides a COS of five outcomes that should be reported in future PGP studies and in clinical practice. It fills an important gap in the literature, as there is currently no COS for pregnancy-related and postpartum-related PGP.

An interesting result of this study is that outcomes in the PGP-COS are "life impact" outcomes. This likely takes consideration of the symptomology of PGP, and can be aligned to the European Guidelines for the diagnosis and treatment of PGP which have a focused PGP treatment aim of relieving pain, improving functional ability and preventing recurrence and chronicity [26]. Additionally, one PGP-COS outcome in particular, fear avoidance, has been heavily studied in low back pain populations, but is not yet extensively studied in pregnancy and postpartum related PGP. Uptake of the PGP-COS will result in increased reporting of fear avoidance in this population and may offer a better understanding of its impact in PGP. Overall, the consensus panel members during discussion expressed that many of the outcomes

**Table 3. Delphi study participant characteristics.**

| Stakeholder Group n (%) | Round 1 (n = 205) | Round 2 (n = 147) | Round 3 (n = 132) |
|---|---|---|---|
| Clinician | 91 (44%) | 59 (40%) | 52 (39%) |
| Clinician Researcher | 38 (19%) | 33 (22%) | 33 (25%) |
| Patient | 42 (20%) | 26 (18%) | 18 (14%) |
| Researcher | 23 (11%) | 21 (14%) | 21 (16%) |
| Service Provider/Policy Maker | 11 (5%) | 8 (5%) | 8 (6%) |
| Gender n (%) | Round 1 (n = 205) | Round 2 (n = 147) | Round 3 (n = 132) |
| Female | 159 (78%) | 114 (78%) | 10 (77%) |
| Male | 45 (22%) | 33 (22%) | 23% |
| Prefer not to say | 1 (0%) | 0 (0%) | 0 (0%) |
| Age n (%) | Round 1 (n = 205) | Round 2 (n = 147) | Round 3 (n = 132) |
| 18–24 | 3 (1%) | 1 (1%) | 1 (1%) |
| 25–34 | 41 (20%) | 28 (19%) | 22 (17%) |
| 35–44 | 74 (36%) | 51 (35%) | 44 (33%) |
| 45–54 | 44 (21%) | 35 (24%) | 33 (25%) |
| 55–64 | 33 (16%) | 26 (18%) | 26 (20%) |
| 65+ | 10 (0%) | 6 (4%) | 6 (5%) |
| Country n (%) | Round 1 (n = 205) | Round 2 (n = 147) | Round 3 (n = 132) |
| Argentina | 1 (0%) | 0 (0%) | 0 (0%) |
| Australia | 8 (4%) | 6 (4%) | 6 (5%) |
| Austria | 1 (0%) | 1 (1%) | 1 (1%) |
| Belgium | 4 (2%) | 4 (3%) | 4 (3%) |
| Brazil | 1 (0%) | 1 (1%) | 1 (1%) |
| Canada | 17 (8%) | 13 (9%) | 13 (10%) |
| Colombia | 1 (0%) | 1 (1%) | 1 (1%) |
| Cook Islands | 1 (0%) | 0 (0%) | 0 (0%) |
| Croatia | 1 (0%) | 0 (0%) | 0 (0%) |
| Denmark | 3 (1%) | 3 (2%) | 3 (2%) |
| Egypt | 1 (0%) | 0 (0%) | 0 (0%) |
| Finland | 1 (0%) | 1 (1%) | 1 (1%) |
| Germany | 1 (0%) | 1 (1%) | 1 (1%) |
| Iceland | 1 (0%) | 0 (0%) | 0 (0%) |
| Iran | 2 (1%) | 2 (1%) | 2 (2%) |
| Ireland | 45 (22%) | 31 (21%) | 24 (18%) |
| Israel | 1 (0%) | 1 (1%) | 1 (1%) |
| Malaysia | 4 (2%) | 2 (1%) | 2 (2%) |
| Mexico | 3 (1%) | 3 (2%) | 2 (2%) |
| Nepal | 2 (1%) | 2 (1%) | 2 (2%) |
| Netherlands | 4 (2%) | 4 (3%) | 4 (3%) |
| New Zealand | 3 (1%) | 2 (1%) | 2 (2%) |
| Norway | 14 (7%) | 11 (7%) | 11 (7%) |
| Philippines | 1 (0%) | 0 (0%) | 0 (0%) |
| Poland | 1 (0%) | 1 (1%) | 1 (1%) |
| Portugal | 3 (1%) | 2 (1%) | 2 (2%) |
| South Africa | 1 (0%) | 1 (1%) | 1 (1%) |
| Sweden | 26 (13%) | 18 (12%) | 18 (14%) |
| Switzerland | 4 (2%) | 4 (3%) | 3 (2%) |
| UK | 28 (14%) | 17 (12%) | 12 (9%) |
| USA | 20 (10%) | 14 (10%) | 13 (10%) |
| Zimbabwe | 1 (0%) | 1 (1%) | 1 (1%) |

**Table 4. Preliminary PGP-COS and consensus meeting voting results.**

| Domain | Outcome | Yes | No |
|---|---|---|---|
| Life impact outcomes *(n = 20)* | **Pain frequency*** | 8 (73%) | 3 (27%) |
| | **Pain intensity/severity*** | 9 (82%) | 2 (18%) |
| | Pain location | 3 (27%) | 8 (73%) |
| | Pain duration/pattern | 3 (27%) | 8 (73%) |
| | **Function/disability/activity limitation*** | 11 (100%) | 0 (0%) |
| | Physical activity levels/exercise limitations | 2 (18%) | 9 (82%) |
| | Sexual functioning | 3 (27%) | 8 (73%) |
| | **Health related quality of life*** | 10 (91%) | 1 (9%) |
| | Health status | 1 (9%) | 10 (91%) |
| | Family life impact | 2 (18%) | 9 (82%) |
| | Social life impact | 2 (18%) | 9 (82%) |
| | Patient satisfaction with life | 2 (18%) | 9 (82%) |
| | Patient satisfaction with treatment | 2 (18%) | 9 (82%) |
| | Anxiety | 1 (9%) | 10 (91%) |
| | Depression | 0 (0%) | 11 (100%) |
| | Emotional symptoms | 1 (9%) | 10 (91%) |
| | **Fear avoidance*** | 8 (73%) | 3 (27%) |
| | Pain catastrophizing | 3 (27%) | 8 (73%) |
| | Self-efficacy | 1 (9%) | 10 (91%) |
| | Sleep function | 4 (36%) | 7 (64%) |
| Resource-use/Economic Impact Outcomes *(n = 1)* | Work ability | 2 (18%) | 9 (82%) |
| Pathophysiological Manifestations/ Clinical Tests Outcomes *(n = 4)* | Gait endurance | 1 (9%) | 10 (91%) |
| | Recovery of symptoms | 3 (27%) | 8 (73%) |
| | Urinary Incontinence | 0 (0%) | 11 (100%) |
| | Motor control/movement strategies/movement patterns | 0 (0%) | 11 (100%) |

* Outcome included in final PGP-COS.

presented in the preliminary PGP-COS were similar or often related and could be captured by the outcomes that were included in the final PGP-COS; in this sense, they considered the five PGP-COS outcomes as critical for measuring and reporting, while also sufficiently broad to effectively assess PGP treatment. Overall, due to its brevity, reporting the outcomes of the PGP-COS, we believe, can be readily assimilated into future research and clinical practice protocols.

## Strengths and limitations

In conducting this study, a number of strengths and limitations are acknowledged. Robust consensus methodology was used to develop the PGP-COS [21]. Additionally, a detailed study protocol was published, prospectively, and the results of this study were presented using the COS-STAR statement guidance to ensure clarity and a high standard of reporting of the PGP-COS [18, 20, 21]. Finally, online Delphi methods were used to capture the views and expertise of an international, multidisciplinary and multi-stakeholder cohort, which also include patient representatives.

This study, however, is not without limitations. Although we sought international participation, our Delphi surveys were created only in English due to time and budgetary constraints. We did try to account for this by using Google Forms to host the surveys, as it provides an

**Table 5. Consensus meeting participant characteristics.**

| Stakeholder Group Representative | n |
|---|---|
| Clinician | 3 |
| Clinician/researcher | 4 |
| Patient | 1 |
| Researcher | 3 |
| Service provider/policy maker | 0 |
| Multiple Stakeholder Group Affiliations | n |
| Patient/clinician | 1 |
| Patient/researcher | 2 |
| Patient/researcher/clinician | 1 |
| Patient/clinician/service provider | 1 |
| Clinician/service provider | 1 |
| Clinician/researcher/service provider | 1 |
| Gender | n |
| Female | 10 |
| Male | 1 |
| Age | n |
| 25–34 | 2 |
| 35–44 | 4 |
| 45–54 | 2 |
| 55–64 | 3 |
| Country of Residence | n |
| Australia | 1 |
| Belgium | 1 |
| Colombia | 1 |
| Ireland | 2 |
| Israel | 1 |
| New Zealand | 1 |
| Norway | 2 |
| Sweden | 1 |
| USA | 1 |
| Profession | n |
| Assistant Professor Musculoskeletal Rehabilitation | 1 |
| Chiropractor | 1 |
| Director of Research | 1 |
| Lecturer Physiotherapy | 1 |
| Manual therapist (Naprapath) | 1 |
| Manual therapist/physiotherapist | 1 |
| Physiotherapist | 2 |
| Physiotherapist/lecturer | 1 |
| Researcher/chiropractor | 1 |
| Unemployed due to PGP | 1 |

option to translate the survey into any language, supported by Google Translate, when viewing in a Google Chrome browser. However, Delphi participants using an alternative web browser would not have had this option. Although, it is plausible that English language surveys could have deterred initial Delphi participation or impacted retention from non-native English-speaking respondents between rounds. In addition to language barriers, lower than expected

uptake in certain geographical regions may also be attributed to less awareness about the condition itself. While we did provide a definition for PGP in the PIL associated with our Delphi survey, differences in individual and provider knowledge and lack of access to services for PGP may have been associated with lower uptake in particular regions. Recognising these limitations, however, we did achieve excellent international representation in the Delphi study (32 countries) and in our consensus meeting (9 countries).

A further limitation to our study is participant representation at the consensus meeting. In our protocol, we had aimed to include 20 experts, and although 25 were invited, only 11 took part. While this is fewer than participant numbers reported in other COS development studies, it adheres to the recommendation of a consensus panel size of 5–11 as recommended by Waggoner and colleagues [27] Additionally, it is similar to previously reported COS development studies in which included a face-to-face consensus meeting [21, 28]. We also did not have equal representation of members from the various stakeholder groups; for example, there was no representation for the service provider/policy maker stakeholder group, as only eight Delphi participants met the eligibility criteria for the meeting and all declined to attend. Additionally, two patient representatives did not attend due to last minute travel emergencies. However, while panel members were invited as representatives of his/her primary stakeholder group affiliation, many identified with more than one stakeholder group, including the service provider/policy maker group (Table 5) In this sense, the voice of this group was present in determining the final PGP-COS. Finally, while lack of equal stakeholder group representation may contribute to bias, is important to note that there is currently no explicit guidelines on which stakeholder groups and how many should be present at the consensus meeting [21]. As a result, representation of the stakeholder groups that participated in the Delphi study present during these face-to-face meetings varies across previously developed COS [21] and in some instances, such as in the non-specific low back pain COS, the steering committee made the final decision in place of a face-to-face meeting with Delphi participants [29]. Further methodological research into COS development would assist in composing more explicit guidance.

## Protocol deviation

In accordance with the COS-STAR statement, it is important to note a deviation from our protocol with regards to the prior consensus definition used in this study [20]. In our protocol, we defined our a priori consensus as outcomes scored by ≥70% of participants as "important" and less than 15% of participants scoring an outcome as "not important" [18]. This definition was initially chosen with the aim of 40 participants in each stakeholder group completing the Delphi study. However, after we closed the first round of the Delphi and before we began analysis, we changed our a priori consensus definition to ≥70% participants rating the outcome as "important" by three of the five stakeholder groups, one of which must have been the patient representative group, due to unequal representation in stakeholder group participation. This was decided so that stakeholder groups with more participants would not dominate the results in each Delphi round and that the "voice" of the patient representative group would always be included.

## Future work

Now that a PGP-COS has been identified, future work is needed. In order to combine and compare outcomes across studies, the outcomes, ideally, should be measured in the same way; that is using the same instrument. The next phase of the PGP-COS study (Fig 1: Phase 4) aims to perform a robust systematic review of measurement properties on all measurement instruments used to measure the five COS outcomes [18]. This review is currently on-going using

COSMIN (COnsensus-based Standards for the selection of health Measurement INstruments) for methodological guidance [30–32]. The results from this review will allow for evidence-based recommendations in the selection of the most suitable measurement instrument to be used for consistent measurement and reporting of the PGP-COS. Finally, efforts will be needed to promote and monitor the uptake of the PGP-COS. The final phase of the PGP-COS study (Fig 1: Phase 5) will involve global promotion of the COS. Details of our plan for global promotion are detailed in the protocol [18]. Promotion and use of the PGP-COS will encourage effective monitoring, increase trial efficiency, improve evidence synthesis and reduce research waste to speed up the development and testing of prevention and management strategies worldwide.

## Conclusion

The evaluation of interventions for PGP in women during pregnancy and postpartum is difficult due to inconsistent outcome reporting. This is the first study to define a COS in this area which can be used in research and clinical settings. Uptake of the consensus-based PGP-COS in all trials, non-randomised studies and clinical settings will encourage consistent reporting, effective monitoring, increased trial efficacy, improved evidence synthesis and reduce research waste to speed up the development and testing of interventional strategies, ultimately resulting in the global promotion of the health and well-being of women with PGP worldwide. The next phase of the PGP-COS will determine how to measure the outcomes of the PGP-COS.

## Supporting information

**S1 Table. List of potential PGP-COS outcome measures.**
(DOCX)

**S2 Table. Delphi results.** S2 Table details the % of stakeholder group that rated an outcome as "important" (4+ on the 5-point rating scale survey or 7+ on the 9-point rating scale survey) above in each Delphi round. [±] Outcome in final PGP-COS [*] Group 1 = clinician; 2 = clinician/researcher; 3 = patient; 4 = researcher; 5 = Service provider/policy maker [a] 5PT = participants responded to Delphi surveys using a 5-point rating Scale [b] 9PT = participants responded to Delphi surveys using a 9-point rating Scale.
(DOCX)

## Acknowledgments

We would like to thank all participants of our Delphi study and consensus meeting for dedicating their time to help develop the PGP-COS.

We would like to acknowledge consensus meeting members Adi Amit David, Anne Randi Høidahl, Daniela Aldabe, Dragana Ceprnja, Gabriel Quintero, Katherine A Pohlman, Lotte Janssens, Miriam Gamble, Natalie Michelle Evensen, and Stine Lilje.

## Author Contributions

**Conceptualization:** Valerie Smith, Francesca Wuytack.

**Data curation:** Alexandria Remus, Annelie Gutke, Juan Jose Saldaña Mena, Lena Nilsson Wikmar, Birgitta Öberg, Christina Olsson, Hilde Stendal Robinson, Britt Stuge, Francesca Wuytack.

**Formal analysis:** Alexandria Remus.

**Funding acquisition:** Valerie Smith, Annelie Gutke, Juan Jose Saldaña Mena, Siv Mørkved, Lena Nilsson Wikmar, Birgitta Öberg, Christina Olsson, Hilde Stendal Robinson, Britt Stuge, Francesca Wuytack.

**Investigation:** Alexandria Remus, Valerie Smith, Annelie Gutke, Juan Jose Saldaña Mena, Siv Mørkved, Lena Nilsson Wikmar, Birgitta Öberg, Christina Olsson, Hilde Stendal Robinson, Britt Stuge, Francesca Wuytack.

**Methodology:** Alexandria Remus, Valerie Smith, Annelie Gutke, Francesca Wuytack.

**Project administration:** Alexandria Remus, Valerie Smith, Francesca Wuytack.

**Supervision:** Valerie Smith, Francesca Wuytack.

**Writing – original draft:** Alexandria Remus.

**Writing – review & editing:** Alexandria Remus, Valerie Smith, Annelie Gutke, Juan Jose Saldaña Mena, Siv Mørkved, Lena Nilsson Wikmar, Birgitta Öberg, Christina Olsson, Hilde Stendal Robinson, Britt Stuge, Francesca Wuytack.

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
