## [Decision Letter · Decision Letter 0]

28 Sep 2020

PONE-D-20-18595

A Core Outcome Set for Research and Clinical Practice in Women with Pelvic Girdle Pain: PGP-COS

PLOS ONE

Dear Dr. Remus,

Thank you for submitting your manuscript to PLOS ONE. After careful consideration, we feel that it has merit but does not fully meet PLOS ONE’s publication criteria as it currently stands. Therefore, we invite you to submit a revised version of the manuscript that addresses the points raised during the review process.

Please consider the reviewers' suggestions below,   and provide further information on issues of potential bias in the methods as suggested by Reviewer 1,  and on the problem in the background information as suggested by Reviewer 2.

We look forward to receiving your revised manuscript.

Kind regards,

Kathleen Finlayson

Academic Editor

PLOS ONE

Journal Requirements:

2. Please include in your financial disclosure statement the name of the funders of this study (as well as grant numbers if available). At present, this information is only available in the funding section of your manuscript.

Thank you for stating the following in the Funding Section of your manuscript:

[We have received funding from the Belgian Chiropractic Union Research Fund

(www.chiropraxie.org) and from the European Centre for Chiropractic Research Excellence

(nikkb.dk/eccre). The funders had no role in the study design, data collection, management,

data interpretation, report writing and decision to submit for publication.]

 [The funders had no role in study design, data collection and analysis, decision to publish, or preparation of the manuscript.]

Additional Editor Comments (if provided):

Thank you for your submission. Please see the suggestions from reviewers below,

Reviewers' comments:

Reviewer's Responses to Questions

**Comments to the Author**

1. Is the manuscript technically sound, and do the data support the conclusions?

Reviewer #1: Yes

Reviewer #2: Yes

2. Has the statistical analysis been performed appropriately and rigorously? 

Reviewer #1: Yes

Reviewer #2: I Don't Know

3. Have the authors made all data underlying the findings in their manuscript fully available?

Reviewer #1: Yes

Reviewer #2: Yes

4. Is the manuscript presented in an intelligible fashion and written in standard English?

Reviewer #1: Yes

Reviewer #2: Yes

5. Review Comments to the Author

Reviewer #1: The paper presents results of a Delphi study to establish a Core Outcome Set (COS) for pelvic girdle pain (PGP) in pregnancy and postpartum. There is a lack of standardization in studies on PGP, especially related to the outcome measures selected for use with the pregnant and postpartum population with PGP. This lack of standardization results in disparate results across studies on the population of interest, and consequently, impedes the synthesis and broader analysis of research on PGP in pregnant and postpartum women. This paper has performed a 3-round Delphi study with an in-person consensus meeting involving multiple stakeholders across several countries. Five outcomes were selected by the Delphi group as the proposed PGP-COS: pain frequency, pain intensity/severity, function/disability/activity limitation, health-related quality of life and fear avoidance.

The rationale, methodology, results and implications of the study are presented clearly and logically. The five outcomes selected for the PGP-COS are logical, clinically relevant, feasible and not duplicative. While the constructs for each outcome are related, the overlap is not excessive and is, in fact, complementary when considering the client with PGP as a whole. All refer to the “life impact” domain, rather than outcomes that focus on specific physical impairments, such as muscle strength, flexibility or joint symmetry. This supports the clinical relevance and feasibility of moving forward with the proposed PGP-COS in clinical practice and research. Inclusion of, and further research on, the proposed PGP-COS from an implementation standpoint would add to the current literature on PGP, as the “life impact” of PGP on childbearing women (on a national or global) level is limited.

One of the biggest challenges in PGP research is the construct of patients’ own expectations of pain during pregnancy and their perceptions of “normal” recovery during the postpartum. While the authors described the online survey process of the first round of the Delphi study, several elements that speak to the constructs of pain, specifically PGP, women’s own expectations, and the cultural aspects of both pain and what is considered “normal” expectations of musculoskeletal pain and functional status in pregnancy and postpartum across countries were not clearly described. This is the biggest challenge to address in this manuscript, as it appears that assumptions may have been made about stakeholders’ shared understanding of PGP and the impact of cultural differences. While the ranking process addressed stakeholders’ shared value of importance for each outcome, it does not address cultural differences or stakeholders’ expectations. These differences or discrepancies in stakeholders’ shared understanding of PGP, expectations and cultural differences may contribute biases to the results of the Delphi study.

The presence and targeted inclusion of international stakeholders is a strength of this Delphi study. The authors have addressed study limitations of using English only survey tools and having fewer participants than projected involved the final in-person consensus meeting. Additional sources of potential bias and limitations in the generalizability of the proposed PGP-COS, as a result, should be considered. Participants were predominantly: clinicians and clinical researchers, female, middle age, and from Ireland, Sweden, USA and Canada. Information on countries’ maternal health systems, specifically public and health provider knowledge of PGP, as well as reimbursement and access to physiotherapy services for PGP in pregnancy and postpartum, may identify important characteristics of stakeholder countries who responded to the Delphi survey compared to those who did not. A narrative of the authors’ perspectives on the characteristics of participants and how those characteristics are representative of the larger community of interest would add to the manuscript.

The link proposed between development of the PGP-COS and evaluating “preventative” strategies for PGP during pregnancy and postpartum is unclear. The PGP-COS will measure “life impact” of PGP, in patients who have already or are currently experiencing PGP in pregnancy or postpartum. Additional description of the link between use of the PGP-COS and preventative strategies is needed.

Reviewer #2: This manuscript describes a diligently methodical process of identifying a well-vetted core outcome set for prenatal and postpartum pelvic girdle pain. I found the manuscript to be well written and to describe a clinically sound process.

One improvement that could be made would be references in the introduction to specific ways in which lack of a COS for PGP has impacted the literature on this problem. For those of us who do not regularly evaluate or manage PGP, this might be helpful.

6. PLOS authors have the option to publish the peer review history of their article (what does this mean?). If published, this will include your full peer review and any attached files.

Reviewer #1: **Yes: **Adrienne H. Simonds, PT, PhD

Reviewer #2: No

---

## [Author Response · Author response to Decision Letter 0]

28 Oct 2020

Thank you for your thorough summary and complementary words about our manuscript. We have addressed your comments as below.

Reviewer #1: The paper presents results of a Delphi study to establish a Core Outcome Set (COS) for pelvic girdle pain (PGP) in pregnancy and postpartum. There is a lack of standardization in studies on PGP, especially related to the outcome measures selected for use with the pregnant and postpartum population with PGP. This lack of standardization results in disparate results across studies on the population of interest, and consequently, impedes the synthesis and broader analysis of research on PGP in pregnant and postpartum women. This paper has performed a 3-round Delphi study with an in-person consensus meeting involving multiple stakeholders across several countries. Five outcomes were selected by the Delphi group as the proposed PGP-COS: pain frequency, pain intensity/severity, function/disability/activity limitation, health-related quality of life and fear avoidance.

Thank you for a concise summary of our manuscript.

The rationale, methodology, results and implications of the study are presented clearly and logically. The five outcomes selected for the PGP-COS are logical, clinically relevant, feasible and not duplicative. While the constructs for each outcome are related, the overlap is not excessive and is, in fact, complementary when considering the client with PGP as a whole. All refer to the “life impact” domain, rather than outcomes that focus on specific physical impairments, such as muscle strength, flexibility or joint symmetry. This supports the clinical relevance and feasibility of moving forward with the proposed PGP-COS in clinical practice and research. Inclusion of, and further research on, the proposed PGP-COS from an implementation standpoint would add to the current literature on PGP, as the “life impact” of PGP on childbearing women (on a national or global) level is limited.

Thank you for your comments. We agree that implementation and uptake of the proposed PGP-COS will add to the current literature on PGP as the “life impact” of PGP on childbearing women is limited.

One of the biggest challenges in PGP research is the construct of patients’ own expectations of pain during pregnancy and their perceptions of “normal” recovery during the postpartum. While the authors described the online survey process of the first round of the Delphi study, several elements that speak to the constructs of pain, specifically PGP, women’s own expectations, and the cultural aspects of both pain and what is considered “normal” expectations of musculoskeletal pain and functional status in pregnancy and postpartum across countries were not clearly described. This is the biggest challenge to address in this manuscript, as it appears that assumptions may have been made about stakeholders’ shared understanding of PGP and the impact of cultural differences. While the ranking process addressed stakeholders’ shared value of importance for each outcome, it does not address cultural differences or stakeholders’ expectations. These differences or discrepancies in stakeholders’ shared understanding of PGP, expectations and cultural differences may contribute biases to the results of the Delphi study.

Thank you for your comment. We acknowledge that there are country and cultural differences regarding expectations of pain during pregnancy and perceptions of “normal” recovery during the postpartum and note that there are individual differences as well. For example, two patients can have different expectations about recovery from pregnancy related PGP. However, in this phase of our overall study, the objective focus was on ‘what to measure’ rather than ‘how’ to measure, or on variations as a result of country or cultural differences. These aspects may emerge in phase 4 of the project where the focus is on ‘how’ to measure the COS outcomes, and in the context of future individual trials/studies where this COS (hopefully) will be used. We are thus confident, while accepting the important point you raise, that this does not bias the Delphi results, but will have a role in the application of the PGP-COS; a point we will attend to in reporting phase 4 of the project (‘how’ to measure the outcomes in the COS). 

The presence and targeted inclusion of international stakeholders is a strength of this Delphi study. The authors have addressed study limitations of using English only survey tools and having fewer participants than projected involved the final in-person consensus meeting. Additional sources of potential bias and limitations in the generalizability of the proposed PGP-COS, as a result, should be considered. Participants were predominantly: clinicians and clinical researchers, female, middle age, and from Ireland, Sweden, USA and Canada. Information on countries’ maternal health systems, specifically public and health provider knowledge of PGP, as well as reimbursement and access to physiotherapy services for PGP in pregnancy and postpartum, may identify important characteristics of stakeholder countries who responded to the Delphi survey compared to those who did not. A narrative of the authors’ perspectives on the characteristics of participants and how those characteristics are representative of the larger community of interest would add to the manuscript.

Thank you for raising this point. We agree that only language and fewer participants than expected in the consensus meeting are not the only limitations in larger uptake from different nations. Additionally, partial reasons for less uptake in some countries can also be attributed to less awareness about the condition both at individual and healthcare system levels. This is reflective of provider knowledge and access to services. We have addressed this as a further limitation to our study in our manuscript.

The link proposed between development of the PGP-COS and evaluating “preventative” strategies for PGP during pregnancy and postpartum is unclear. The PGP-COS will measure “life impact” of PGP, in patients who have already or are currently experiencing PGP in pregnancy or postpartum. Additional description of the link between use of the PGP-COS and preventative strategies is needed.

Thank you. For clarity we have removed references to “preventative strategies” and refer now to a COS for interventions for PGP. We have slightly tweaked the ‘Aim’ also to clarify this further, which now reads; The aim of our study was to develop a consensus-based COS for PGP during pregnancy and postpartum that should be used, as a minimum, for research and clinical practice.

Reviewer #2: This manuscript describes a diligently methodical process of identifying a well-vetted core outcome set for prenatal and postpartum pelvic girdle pain. I found the manuscript to be well written and to describe a clinically sound process.

One improvement that could be made would be references in the introduction to specific ways in which lack of a COS for PGP has impacted the literature on this problem. For those of us who do not regularly evaluate or manage PGP, this might be helpful.

Thank you for your time in reviewing our manuscript. We have added specific comments to the introduction (paragraph 1) in which the lack of a PGP COS has impacted the literature for clarity to those who do not regularly evaluate or manage PGP.

---

## [Decision Letter · Decision Letter 1]

15 Dec 2020

PONE-D-20-18595R1

A Core Outcome Set for Research and Clinical Practice in Women with Pelvic Girdle Pain: PGP-COS

PLOS ONE

Dear Dr. Remus,

Thank you for submitting your manuscript to PLOS ONE. After careful consideration, we feel that it has merit but does not fully meet PLOS ONE’s publication criteria as it currently stands. Therefore, we invite you to submit a revised version of the manuscript that addresses the points raised during the review process.

Thank you for addressing the reviewers' comments.   Please see and consider further reviewer suggestions below,  in particular,  including "comparison of the round 3 stakeholders' response, or reporting round 3 results and presenting how they compare to the final 11 participants' consensus opinions", to address the issue discussed.

We look forward to receiving your revised manuscript.

Kind regards,

Kathleen Finlayson

Academic Editor

PLOS ONE

Reviewers' comments:

Reviewer's Responses to Questions

**Comments to the Author**

1. If the authors have adequately addressed your comments raised in a previous round of review and you feel that this manuscript is now acceptable for publication, you may indicate that here to bypass the “Comments to the Author” section, enter your conflict of interest statement in the “Confidential to Editor” section, and submit your "Accept" recommendation.

Reviewer #1: (No Response)

Reviewer #2: All comments have been addressed

2. Is the manuscript technically sound, and do the data support the conclusions?

Reviewer #1: Yes

Reviewer #2: Yes

3. Has the statistical analysis been performed appropriately and rigorously? 

Reviewer #1: Yes

Reviewer #2: N/A

4. Have the authors made all data underlying the findings in their manuscript fully available?

Reviewer #1: Yes

Reviewer #2: Yes

5. Is the manuscript presented in an intelligible fashion and written in standard English?

Reviewer #1: Yes

Reviewer #2: Yes

6. Review Comments to the Author

Reviewer #1: The authors have addressed prior feedback early on in the paper well. Minor edits are suggested below.

Inconsistencies noted in the organization of preliminary outcomes (Table 1), with some lack of clarity on the transition between rounds.

1. The "Pathophysiological Manifestations" title in column 3 may also include aspects of pain, depression and anxiety, sleep disturbance, emotional symptoms, confidence and wellbeing, etc. These outcomes are already listed in "Life Impact". There appears to be some redundancy across those 2 columns, and in categorizing outcomes, the authors might consider including "Pathophysiological Manifestations" into "Life Impact" and leave the clinical tests in this freestanding category.

2. The outcomes included in the third column - "Pathophysiological Manifestations/Clinical Tests" of maternal pregnancy outcome, newborn outcome and surgical outcome do not seem to be appropriately categorized. Perhaps these should move to "Adverse Events and/or Medical Outcomes" to capture these factors, or be assigned a stand-alone column for Medical Outcomes.

3. The "anthropometric outcome" is unclear - does this refer to weight gain and/or BMI during pregnancy?

4. "Recovery of symptoms" seems too broad. This may then include "full pain recovery" which is listed in "Life Impact". These categories should be mutually exclusive, meaning that outcomes listed across may be related, but cannot be interchanged for each other. Perhaps some recoding of categories and/or re-evaluation of outcomes is needed.

Rating scales used to assess preliminary outcomes in Round 1 and 2 of the Delphi study include both a 5-point and a 9-point scale, administered randomly. Explanation of the rationale for use of the 2 different scales is suggested. Although it is mentioned this will be presented in a separate paper, it is a point of confusion for this study's methodology.

The limitations of the consensus meeting are again, striking. To base the final round of inclusion on the opinions of 11 individuals, when 132 stakeholders completed round 3, seems potentially biased. This final consensus round, arguably the most important and most impactful round on results of this Delphi study, represented the opinions of only 17% of the stakeholders. While this limitation is explained because of travel emergencies, it is suggested the authors perform additional statistical analyses on the results of round 3 and/or explore a remote voting process for the 83% of round 3 stakeholders whose voices and opinions were not heard in this final round of the Delphi process. Perhaps a comparison of the round 3 stakeholders' response, or a virtual consensus meeting may be needed, or reporting round 3 results and presenting how they compare to the final 11 participants' consensus opinions.

Additionally, of the 11 participants at the consensus meeting, 6 (54.5%) were researchers. This also contributes to bias. While the authors disclose participants at the consensus meeting holding multiple roles, this point should be made explicit.

Reviewer #2: My concerns were limited and have been satisfactorily addressed. I have not other concerns at that this time.

7. PLOS authors have the option to publish the peer review history of their article (what does this mean?). If published, this will include your full peer review and any attached files.

Reviewer #1: **Yes: **Adrienne H. Simonds, PT, PhD

Reviewer #2: No

---

## [Author Response · Author response to Decision Letter 1]

29 Jan 2021

Dear Editor and Reviewers, 

Thank you for your helpful comments on our revised paper. We have attended to the comments with point by point responses as below, and in further revisions to the manuscript. We hope these meet the requirements for progressing our paper to publication.

Editor:

Thank you for addressing the reviewers' comments. Please see and consider further reviewer suggestions below, in particular, including "comparison of the round 3 stakeholders' response, or reporting round 3 results and presenting how they compare to the final 11 participants' consensus opinions", to address the issue discussed.

Thank you. We have addressed this comment of reviewer 1 below. While direct statistical comparison of round 3 responses with the results of the consensus meeting was not conducted since it was not feasible nor appropriate (see reasons in response to reviewer’s comment), readers can easily compare the round 3 and consensus meeting results from the information provided in table 4 and supplementary table 2. This is now also mentioned in the manuscript. 

Reviewer #1: The authors have addressed prior feedback early on in the paper well. Minor edits are suggested below.

Inconsistencies noted in the organization of preliminary outcomes (Table 1), with some lack of clarity on the transition between rounds.

1. The "Pathophysiological Manifestations" title in column 3 may also include aspects of pain, depression and anxiety, sleep disturbance, emotional symptoms, confidence and wellbeing, etc. These outcomes are already listed in "Life Impact". There appears to be some redundancy across those 2 columns, and in categorizing outcomes, the authors might consider including "Pathophysiological Manifestations" into "Life Impact" and leave the clinical tests in this freestanding category.

Thank you for your comment. Our outcome categorisation and subsequent table headings adhere to the OMERACT 2.0 Filter Framework (Boers et al. 2014). In this framework Life Impact and Pathophysiological Manifestations are established as separate domains. According to this framework, aspects of pain, depression and anxiety, sleep disturbance, emotional symptoms, confidence and wellbeing, etc. are classified as Life Impact domain outcomes (Fig 1; Boers et al 2014). We have kept Table 1 column titles as is for consistency with this framework and also the previous publication in which possible outcomes for the PGP-COS were identified (Wuytack and O’Donovan. 2019). 

2. The outcomes included in the third column - "Pathophysiological Manifestations/Clinical Tests" of maternal pregnancy outcome, newborn outcome and surgical outcome do not seem to be appropriately categorized. Perhaps these should move to "Adverse Events and/or Medical Outcomes" to capture these factors, or be assigned a stand-alone column for Medical Outcomes.

Thank you for your comment. As described above, the outcomes are categorised according to the OMERACT 2.0 filter framework (Boers et al. 2014) and our previous publication identifying potential outcomes for the Delphi study (Wuytack and O’Donovan. 2019). We have kept categorisation as is for consistency as well as the outcomes were grouped as such during the Delphi Study. We have included a statement addressing this in the main text (page 6).

3. The "anthropometric outcome" is unclear - does this refer to weight gain and/or BMI during pregnancy?

Thank you for your comment. Anthropometric outcomes were identified in our previous systematic review (Wuytack and O’Donovan. 2019) as well as in Supplementary Table 1 of this manuscript. Anthropometric outcomes were defined as measurements of size and proportion of the human body for example, height, weight, BMI, etc.. Reference to S1 Table is made in the Delphi Procedure section of methods. It is not referenced in Table 1 as this is just a list of the potential outcomes.

4. "Recovery of symptoms" seems too broad. This may then include "full pain recovery" which is listed in "Life Impact". These categories should be mutually exclusive, meaning that outcomes listed across may be related, but cannot be interchanged for each other. Perhaps some recoding of categories and/or re-evaluation of outcomes is needed.

Thank you comment. Full pain recovery in our Delphi survey was defined as “being pain free/no longer in pain after treatment.” This is separate to recovery of symptoms as symptoms can include more than pain. The definition is also included in Supplementary Table 1. As mentioned above we have kept categorisation as is for consistency across publications as well as this is how outcomes were identified in studies included the systematic review leading into the Delphi (Wuytack & O’Donovan 2019) and were presented to participants during the Delphi Study.

Rating scales used to assess preliminary outcomes in Round 1 and 2 of the Delphi study include both a 5-point and a 9-point scale, administered randomly. Explanation of the rationale for use of the 2 different scales is suggested. Although it is mentioned this will be presented in a separate paper, it is a point of confusion for this study's methodology.

Thank you for your comment. We have added an explanation of our rational for using the two scales in the main text. The findings from the embedded methodological study have since been published and we have addressed the text appropriately for reader’s reference (page 9).

The limitations of the consensus meeting are again, striking. To base the final round of inclusion on the opinions of 11 individuals, when 132 stakeholders completed round 3, seems potentially biased. This final consensus round, arguably the most important and most impactful round on results of this Delphi study, represented the opinions of only 17% of the stakeholders. While this limitation is explained because of travel emergencies, it is suggested the authors perform additional statistical analyses on the results of round 3 and/or explore a remote voting process for the 83% of round 3 stakeholders whose voices and opinions were not heard in this final round of the Delphi process. Perhaps a comparison of the round 3 stakeholders' response, or a virtual consensus meeting may be needed, or reporting round 3 results and presenting how they compare to the final 11 participants' consensus opinions.

Thank you for your comment. We understand your concern regarding 11 participants in our face-to-face consensus meeting. However, we adhered to the current methodological guidelines for Core Outcome Set development developed by COMET (Core Outcome Measures in Effectiveness Trials Initiative). According to these guidelines, the face-to-face consensus meeting includes a subset of participants. These guidelines do not include a required number of participants. However, Waggoner, Carline and Durning. 2016 have previously explored the optimal consensus panel size and have determined 5-11 members is recommended. We had referenced this in our limitations but since realised that the article was incorrectly referenced. This has been rectified and is now included in our reference list. In line with these recommendations, most other previous COS development projects use a similar size sub-group of Delphi participants for the face-to-face. It is also important to note that while the COMET handbook does recommend a face-to-face meeting, they also refer to a review that identified that not all COS include this step in their development (page 24 of the handbook). While the face-to-face meeting may seem to be the most important and most impactful round, the three previous rounds of the Delphi study are just as important. The Delphi process is a consensus process itself. 

In our Supplemental Table 2 (S2 Table) we have provided how all stakeholder groups voted throughout all rounds and have also highlighted the final five outcomes included in the PGP-COS. From this table, comparisons between the consensus meeting and round 3 responses can be made. However, statistical comparison of round 3 responses with the results of the consensus meeting was not feasible nor appropriate due to the following reasons: 1) This is not recommended in current COS development methodology (COMET handbook); 2) The consensus definition in the Delphi and face-to-face meeting were different in that for the Delphi >70% of only 3 stakeholder groups (including at least the patient group) needed to rate an outcome as important, while for the meeting >70% of all participants had to vote an outcome as important (This is line with current COS development guidelines; 3) The 2 different scales we used makes for the embedded methodological study. For example 6 outcomes that were voted on in the consensus meeting were only voted on by one of the scale groups (not both). 

Some outcomes that were rated high by all stakeholder groups in Round 3 were not subsequently included in the final COS during the face-to-face consensus meeting. However, this is expected during the consensus process aiming at identifying a small number of core outcomes that should and could be measured in all trials and in clinical practice. As more and more rounds are incorporated during the whole consensus process, more and more outcomes begin to not reach consensus. This is evidenced by starting with 53 outcomes in Round 1 to only 20 outcomes reaching consensus in round 3. It is therefore expected that the number of outcomes reaching consensus in round 3 would reduce during the final face-to-face meeting. Several outcomes from round 3 (e.g. sexual functioning) were also not included in the final COS because it was agreed by the participants that they formed part of other outcomes that were included in the final COS (e.g. activity limitations) (this is mentioned in the discussion on page 16). Finally, it is also important to note that only 13% of Delphi participants expressed interest in participating in the face-to-face consensus meeting when asked at completion of the survey (being informed that travel and accomodation costs would be funded). As such, it would not be feasible or methodologically appropriate to include all participants from round 3 in the face-to-face consensus meeting.

Additionally, of the 11 participants at the consensus meeting, 6 (54.5%) were researchers. This also contributes to bias. While the authors disclose participants at the consensus meeting holding multiple roles, this point should be made explicit.

Thank you for your comment. Table 5 reports that 3 participants were researchers, 3 participants were clinicians and 4 participants were clinician/researchers. While having a larger number of researchers may contribute to bias, there is currently no consensus on who should be involved in the face-to-face meeting (COMET Handbook). Table 5 also reports the multiple roles by the 11 participants (including participants who were patients in addition to belonging to other stakeholder group(s)). We have addressed the text to be more explicit about the multiple roles of these participants (page 14 and 18).

---

## [Editor Report · Decision Letter 2]

8 Feb 2021

A Core Outcome Set for Research and Clinical Practice in Women with Pelvic Girdle Pain: PGP-COS

PONE-D-20-18595R2

Dear Dr. Remus,

We’re pleased to inform you that your manuscript has been judged scientifically suitable for publication and will be formally accepted for publication once it meets all outstanding technical requirements.

Kind regards,

Kathleen Finlayson

Academic Editor

PLOS ONE

Additional Editor Comments (optional):

Thank you for addressing the reviewers' comments and clarifying the queries on methods.

Please consider adding an edit to address the query about 'anthropometric outcome' definition in Table 1 - (e.g. add BMI in brackets?, or the reference)
---

## [Editor Report · Acceptance letter]

16 Feb 2021

PONE-D-20-18595R2 

A Core Outcome Set for Research and Clinical Practice in Women with Pelvic Girdle Pain: PGP-COS 

Dear Dr. Remus:

I'm pleased to inform you that your manuscript has been deemed suitable for publication in PLOS ONE. Congratulations! Your manuscript is now with our production department. 

Kind regards, 

on behalf of

Dr. Kathleen Finlayson 

Academic Editor

PLOS ONE